# Effects of Canopy Nitrogen Addition and Understory Vegetation Removal on Nitrogen Transformations in a Subtropical Forest

Saif Ullah [1,2], Wenfei Liu [3], Jawad Ali Shah [1,2], Fangfang Shen [3], Yingchun Liao [3], Honglang Duan [4], Guomin Huang [3] and Jianping Wu [1,2,*]

1 Ministry of Education Key Laboratory for Transboundary Ecosecurity of Southwest China, Yunnan Key Laboratory of Plant Reproductive Adaptation and Evolutionary Ecology and Institute of Biodiversity, School of Ecology and Environmental Science, Yunnan University, Kunming 650500, China; saif.haryankot1@outlook.com (S.U.); ali_jawad009@yahoo.com (J.A.S.)

2 Key Laboratory of Soil Ecology and Health in Universities of Yunnan Province, Yunnan University, Kunming 650500, China

3 Jiangxi Key Laboratory for Restoration of Degraded Ecosystems & Watershed Ecohydrology, Nanchang Institute of Technology, Nanchang 330099, China; liuwf729@126.com (W.L.); shenfangfang@nit.edu.cn (F.S.); liaoyingc@163.com (Y.L.); g.huang@nit.edu.cn (G.H.)

4 Key Laboratory of Forest Cultivation in Plateau Mountain of Guizhou Province, Institute for Forest Resources and Environment of Guizhou, College of Forestry, Guizhou University, Guiyang 550025, China; hlduan@gzu.edu.cn

* Correspondence: jianping.wu@ynu.edu.cn

**Abstract:** The management of understory vegetation and anthropogenic nitrogen (N) deposition has significantly resulted in a nutrient imbalance in forest ecosystems. However, the effects of canopy nitrogen addition and understory vegetation removal on N transformation processes (mineralization, nitrification, ammonification, and leaching) along with seasonal variations (spring, summer, autumn, and winter) remain unclear in subtropical forests. To fill this research gap, a field manipulation experiment was conducted with four treatments, including: (i) CK, control; (ii) CN, canopy nitrogen addition (25 kg N ha$^{-1}$ year$^{-1}$); (iii) UR, understory vegetation removal; and (iv) CN+UR, canopy nitrogen addition plus understory vegetation removal. The results revealed that CN increased net mineralization and nitrification by 294 mg N m$^{-2}$ month$^{-1}$ in the spring and 126 mg N m$^{-2}$ month$^{-1}$ in the winter, respectively. UR increased N mineralization and nitrification rates by 618 mg N m$^{-2}$ month$^{-1}$ in the summer. In addition, CN effectively reduced N leaching in the spring, winter, and autumn, while UR increased it in the spring and winter. UR increased annual nitrification rates by 93.4%, 90.3%, and 38.9% in the winter, spring, and summer, respectively. Additionally, both net N ammonification and annual nitrification rates responded positively to phosphorus availability during the autumn. Overall, UR potentially boosted nitrification rates in the summer and ammonification in the spring and winter, while CN reduced N leaching in the spring, winter, and autumn. Future research should integrate canopy nitrogen addition, understory vegetation removal, and phosphorus availability to address the global N deposition challenges in forest ecosystems.

**Keywords:** ammonification; available phosphorus; nitrogen deposition; nitrogen mineralization; seasonality

## 1. Introduction

The anthropogenic N deposition in forest ecosystems poses a significant threat to sensitive ecological functions and processes in a wide range of regions [1,2]. Atmospheric N deposition has increased significantly over the past two decades and is expected to increase further in the near future [3,4]. On the one hand, an increased influx of atmospheric N can greatly promote plant growth and increase N levels in forests [5]. On the other hand, this situation raises significant environmental concerns, as it negatively affects the soil fertility and ecosystem processes. For instance, the excessive application of N can also

lead to soil acidification, nutrient imbalances, and plant damage [6,7]. Furthermore, the addition of N has significantly increased nitrification, leading to the saturation of N and causing remarkable changes in crucial N cycling processes in forest ecosystems [8,9]. As such, N-saturated ecosystems have higher rates of N mineralization, nitrification, and N losses through the leaching of soil nitrate [7,10]. The dynamics of soil N status and the associated processes have significant implications for various changes in the ecosystem structure and function [11]. Therefore, it is essential to understand the status of N in the soil and the changes in response to N deposition to ensure sustainable forest management practices for upcoming global ecological concerns.

The imbalance between N availability and demand during seasonal shifts in the forest ecosystem can profoundly affect N cycling processes. Studies conducted in eastern hardwood forests have suggested a direct link between the rise in N losses during the non-growing season and the decline in plant N uptake [12,13]. Over time, ecologists have widely accepted the notion of the "growing season", assuming it coincides with the plant N uptake and demand [14]. However, this assumption has not been thoroughly examined, particularly in subtropical forests, where the influence of seasonal transitions on both the net and annual N cycles has been largely ignored.

Numerous studies have shown that the rate of N transformation processes, such as mineralization, nitrification, ammonification, and the leaching of nitrates, mainly governs the availability of inorganic N in the soil. The canopy nitrogen addition (CN) and understory vegetation removal (UR) significantly regulate the soil nutrient availability [15,16]. Disrupting the close connections between the overstory and understory vegetation has significant implications for soil N pools and transformations in forest ecosystems [5,17]. Tree canopy litterfall contributes about half of the N in the soil, which plants can reuse, and affects the available phosphorus (AP) of the soil, pH, and litter decomposition, thus affecting the cycling of N in the soil [18,19]. AP can influence the soil N pools and cycling mechanisms through various effects on plant growth and microbial activity. For example, an increase in AP leads to reduced N leaching losses and enhanced N turnover rates within plant soil ecosystems by stimulating the growth of plants and microbes and facilitating N uptake [20,21]. The literature has shown that UR can increase or decrease soil N turnover rates, with the effects varying by ecosystem type, experimental duration, and environmental factors [22,23]. Furthermore, the N cycle is influenced by various abiotic and biotic factors, including soil pH, N addition, moisture, temperature, carbon, microbial biomass, nitrogen, C:N ratio, and the availability of labile C [24,25]. Although many studies have investigated N cycling processes in N-limited forest ecosystems [26,27], our understanding of the effects of CN, UR, and seasonality on soil N processes in subtropical forests is still limited.

In previous research, conventional N fertilization experiments were conducted to provide N to the understory or forest floor, ignoring the potential uptake of N by leaves [28]. However, the credibility of these findings has come under scrutiny as they do not consider the canopy processes, such as interception and assimilation [29]. In forest ecosystems, most of the reactive N that is deposited goes through the tree canopy first. During this process, the tree canopy can absorb it through the leaves or immobilize it with the help of the canopy biota before the remaining N reaches the forest floor [30,31]. Previous studies have shown that when 25 kg N ha$^{-1}$ year$^{-1}$ was added to the canopy, 52% of the N was captured in the canopy, and when 50 kg N ha$^{-1}$ year$^{-1}$ was added, 44% of the N was intercepted [32]. Most of the N intercepted was taken up by leaves and stored in plants, suggesting that adding N to the canopy can promote tree growth and ecosystem functions [33]. The present study has significant value in terms of its potential applicability in understanding the status of N and the dynamics in soil, especially in terms of an appropriate method to increase atmospheric N deposition in the forest ecosystem. To date, there have been rare comprehensive studies using CN, UR, and seasonality approaches to examine the effects of N deposition on soil N status and dynamics.

Both regionally and globally, the health of forest ecosystems is threatened by the increase in atmospheric N deposition. Atmospheric N input is considered an important source of nutrients for plant growth in an N-limited ecosystem; however, it remains an open question regarding how elevated N deposition affects plant growth in subtropical forest ecosystems. The objective of this study was to evaluate the responses to soil N dynamics in subtropical forests, considering CN, UR, and CN + UR (both canopy nitrogen addition and understory vegetation removal), and a controlled plot (CK). The experiment was conducted at the Guanzhuang National Forestry Farm in the Fujian province, south China, in a subtropical forest. The study area was known to have significant atmospheric N deposition [34]. In this study, net N mineralization, nitrification, ammonification, and N leaching rates were measured after 2 years of experimental treatments with CN and UR. To investigate the effects of CN and UR on N transformation and leaching rates, measurements were taken in the mineral soil layer (0–10 cm) during the four seasons in this study: spring, summer, autumn, and winter. In particular, the study investigated the following research objectives: (i) Evaluate the impact of CN and UR on nitrogen transformation rates in subtropical forest ecosystems, especially considering their effects on the net and annual nitrogen rates within the different seasons; (ii) Analyze the influence of CN, UR, and seasonal variations on nitrate leaching dynamics within forest ecosystems; (iii) Investigate the response of nitrogen transformation rates to fluctuations in the soil water content, available phosphorus, and pH after canopy nitrogen addition and understory vegetation removal.

## 2. Materials and Methods

### 2.1. Study Site

The experimental site was located in Sanming city, at the Guanzhuang National Forestry Farm (26°30 N and 117°43 E), in the Fujian province, southern China. The ecosystem had a typical subtropical monsoon climate with a mean annual precipitation range of 1606–1650 mm and a mean annual temperature range of 18.8–19.6 °C. The spring average temperature was 18.5 °C, with an average precipitation of 220.6 mm; the summer average was 27.9 °C, with an average precipitation of 177.3 mm; the autumn average was 21.3 °C, with an average precipitation of 73.3 mm; and the winter average was 12.7 °C, with an average precipitation of 94.5 mm, covering the period from March 2016 to February 2017. The soil is an acrisol in the US soil taxonomy (soil organic carbon 18.39 g kg$^{-1}$, soil bulk density 1.06 g cm$^{-3}$, and soil pH of 4.68). The study used ongoing N fertilization changes in the *Cunninghamia lanceolata (*Lamb.) Hook. plantation forest that was planted in 2008 with a density of 1660 trees per ha$^{-1}$. During the study period, the average height and diameter at breast height measured 5.11 m and 7.34 cm, respectively. The dominant understory species included *Ardisia punctate* L., *Smilax china* L., *Arachniodes hasseltii* Ching., and *Ficus hirta* Vahl. [27,35].

### 2.2. Experimental Design and Measurements

The randomized block experiment was established in June 2014 when the *C. lanceolata* plantations were 6 years old. Briefly, eight 15 m × 15 m plots were randomly placed on a 6-ha section of the plantation and grouped into four blocks, receiving either canopy nitrogen addition (25 kg N ha$^{-1}$ year$^{-1}$) or control with no alterations; within each plot, a similar block was designated for the removal of the understory. There were two factors with four treatments randomly located in each of the four replicated blocks, as follows: (i) CK, control; (ii) CN, canopy nitrogen addition; (iii) UR, understory removal; and (iv) CN + UR, canopy nitrogen addition plus understory removal (Figure 1). A 3–8 m large buffer strip was used to avoid cross-contamination between the adjacent plots. For each N addition block of the canopy, the required amount of NH$_4$NO$_3$ (269 g) was dissolved in 10 L of tap water, and the solution was evenly sprayed with a sprayer on the forest canopy within the plots every 2 months and continued until the end of the sampling period [27]. To apply the N on the forest canopy, a high-pressure spraying apparatus was placed at the

center of each plot. Essentially, the N solutions were pumped through PVC pipes (10 cm in diameter) attached to a tall support tower, reaching a height of 5 m above the forest canopy. Crane sprinklers were used to evenly spray the N solution over the canopy, rotating 360° for uniform coverage. The control block received an equivalent volume of water without $NH_4NO_3$. For the understory vegetation removal block, the understory plants were manually removed monthly using a machete and a hoe [27].

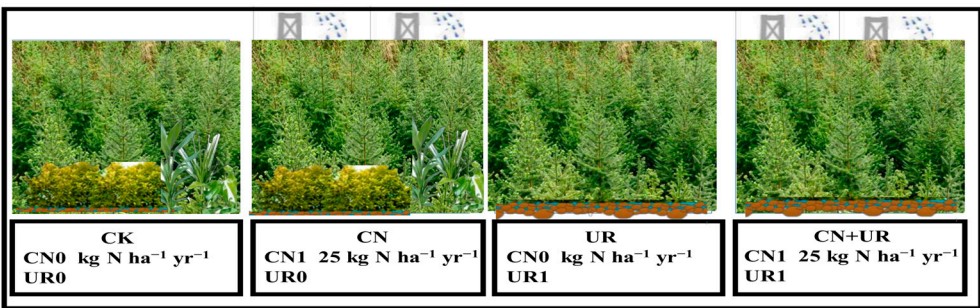

**Figure 1.** Four blocks in each plot represent the conceptual experimental design of this study. CN0 and UR0 represent no canopy N addition or understory removal, while CN1 and UR1 represent the canopy N addition or understory removal treatment, respectively.

The mineralization and nitrification rates were calculated by applying the PVC pipe cover method for soil incubation in situ [36]. In this method, the root system of the plant was cut off and the water and gas exchange was stopped, potentially affecting the N mineralization rate of the soil. This technique is appropriate for estimating N mineralization in forest soil within its natural environment [37]. In April 2016, three points were selected in each quadrate, each having similar basic soil properties. Three polyvinyl chloride tubes (15 cm high, 5 cm in diameter) were placed at a depth of 10 cm on each treatment block. The initial soil sample (S1) from tube 1 was immediately collected. In the case of tube 2, designated as enclosed incubated soil (S2), it was sealed with a lid, but holes were strategically added to the sidewalls, to ensure proper aeration and simulate the changing conditions of N deposition in the surrounding soil. Regarding tube 3, known as open incubated soil (S3), it was left uncovered, allowing the rainwater to freely enter the tube. Soil samples from tube 2 (S2) and tube 3 (S3) were incubated for 3 months in the field before being collected. Similar cycles were replicated three times in July 2016, October 2016, and January 2017. The four incubation periods corresponded to spring, summer, autumn, and winter, respectively.

The soil samples were collected and carefully processed for analysis. After removing litter and stones, the soil was sieved through 2 mm sieves, and the bulk density of the soil was measured using a drying technique. The soil water content was determined using the gravimetric method, measuring grams of water per 100 g of dry soil, achieved by drying the fresh soil to a constant weight at 105 °C. The pH of the soil was evaluated using a mixture of soil and deionized water in a 1:2.5 (*w/v*) ratio. The soil samples were then extracted for mineral N within 48 h after collection and stored at 4 °C for the analysis. The extraction involved using 20 g of fresh soil mixed with 100 mL of a 2 M KCL solution in a 1:5 ratio. Subsequently, the concentrations of ammonium and nitrate were determined using the salicylate–nitroprusside and sulfanilamide methods, respectively, using a flow injection autoanalyzer (FIA, Lachat Instruments, Loveland, CO, USA) [38,39]. Furthermore, the available phosphorus in the soil was extracted using a Bray-2 solution [40] and analyzed using the blue colorimetric method of molybdate.

### 2.3. Nitrogen Mineralization Measurement

In this study, three soil samples were collected from different tubes: tube 1 (S1) was taken out immediately; tube 2 (S2) had a lid but had holes for aeration; and tube 3 (S3) was left open to allow rainwater to enter. The net mineralization rate of N was calculated by

deducting the final amounts of nitrate ($NO_3^-$-N) and ammonium ($NH_4^+$-N) in incubated tube 2 (S2) from their initial amounts (S1). To determine the net nitrification rate, the starting $NO_3^-$-N content in tube 1 (S1) was subtracted from the final $NO_3^-$-N content in tube 2 (S2) [41]. The amount of $NH4^+$-N and $NO_3^-$-N in the enclosed soil core (S2) and the open core (S3) were compared to calculate the amount of N leaching. In our study, we employed the following equations and methods to determine the N conversion rates.

The ammonification and the nitrification rate were calculated as the differences in $NH_4^+$-N and $NO_3^-$-N before and after soil incubation, respectively. The mineralization rate of soil N was calculated using the following formulas:

$$\text{Ammonification rate} = (NH_4^+\text{-N}_{after} - NH_4^+\text{-N}_{before})/t \tag{1}$$

$$\text{Nitrification rate} = (NO_3^-\text{-N}_{after} - NO_3^-\text{-N}_{before})/t \tag{2}$$

$$\text{Net N mineralization rate} = \text{ammonification rate} + \text{nitrification rate}$$

$$\text{Nitrogen leaching rate} = \text{tube 3 (S3)} - \text{tube 2 (S2)}$$

where the subscripts after and before represent the soil $NH_4^+$-N and $NO_3^-$-N concentrations after and before incubation (mg kg$^{-1}$), respectively. t is the incubation time (days).

To convert the N nitrification rates from a monthly to an annual basis, we used the following formula:

$$A = \sum Si \times 4/100 \tag{3}$$

where A represents the annual N nitrification rates (kg N ha$^{-1}$ year$^{-1}$), and Si denotes the monthly N nitrification or mineralization rates (mg N m$^{-2}$ month$^{-1}$) in each season (spring, summer, autumn, and winter). The conversion constants 4 and 1/100 were applied to convert from a season to a year and from mg N m$^{-2}$ to kg N ha$^{-1}$, respectively. We calculated the N nitrification and mineralization rates in each season and then summed them to obtain the annual rates.

*2.4. Statistical Analyses*

Before the statistical analysis, the homogeneity of the data (Levene's test) and its normality (Shapiro–Wilk test) were verified. A one-way analysis of variance (ANOVA) followed by a Tukey HSD post hoc test was employed to find the differences between the treatments applied. We used a three-way analysis of variance (ANOVA) to examine the influence of the treatments (UR, CN, and season) as the primary factor on the dynamics of N processes, as well as on the soil chemical properties, such as available phosphorus, soil water content, and soil pH. The various relationships were evaluated using Pearson's correlations in Sigma plot 12 (Systat Software Inc., Chicago, IL, USA). SPSS 25 (IBM Inc., Chicago, IL, USA) and GraphPad Prism 12.0 (GraphPad Software, Inc., San Diego, CA, USA) were used to perform the statistical analyses and illustrations. Differences were considered significant at a critical level of alpha 0.05.

## 3. Results

*3.1. Soil Physiochemical Properties*

UR had no significant effect on the soil water content, while CN, S, and their interactions significantly impacted SWC (Figure 2b). The soil AP was significantly affected by UR ($p < 0.001$), CN ($p = 0.01$), and season (S) ($p < 0.001$), with significant interactions between the treatments on the soil AP ($p < 0.05$; Figure 2a). There were no statistically significant differences in the soil pH between the various applied treatments (Figure 2c).

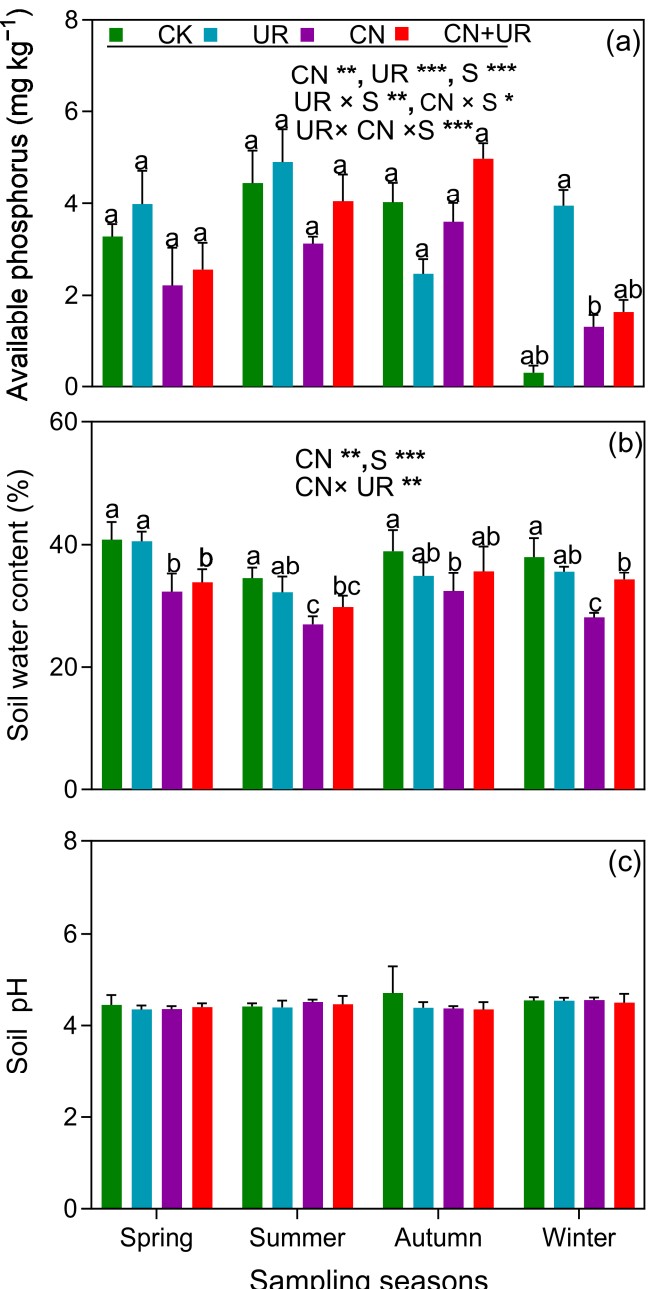

**Figure 2.** Concentrations of available phosphorus (**a**), soil water content (**b**), and soil pH (**c**) under different treatments. Data are presented as means ± SE. CK: control group, UR: understory removal, CN: canopy nitrogen addition, CN + UR: canopy nitrogen addition plus understory removal, S: season. The inserted significant values are derived from three-way ANOVAs. Different lowercase letters represent significant differences among treatments. *, **, and *** indicate the significance level of $p < 0.05$, $p < 0.01$, and $p < 0.001$.

### 3.2. N Transformation under CN and UR

The net mineralization and nitrification rates varied substantially between the seasons, with the highest impact observed during summer, particularly with UR treatment at 618 mg N m$^{-2}$ month$^{-1}$. CN showed rates of 294 mg N m$^{-2}$ month$^{-1}$ in the spring and 126 mg N m$^{-2}$ month$^{-1}$ in the winter. UR consistently displayed the highest values for both net mineralization and nitrification rates during the summer (Figure 3a,b). The interactions between CN and UR, as well as between UR, CN, and the season (S), significantly affected the in situ rates of net mineralization and nitrification.

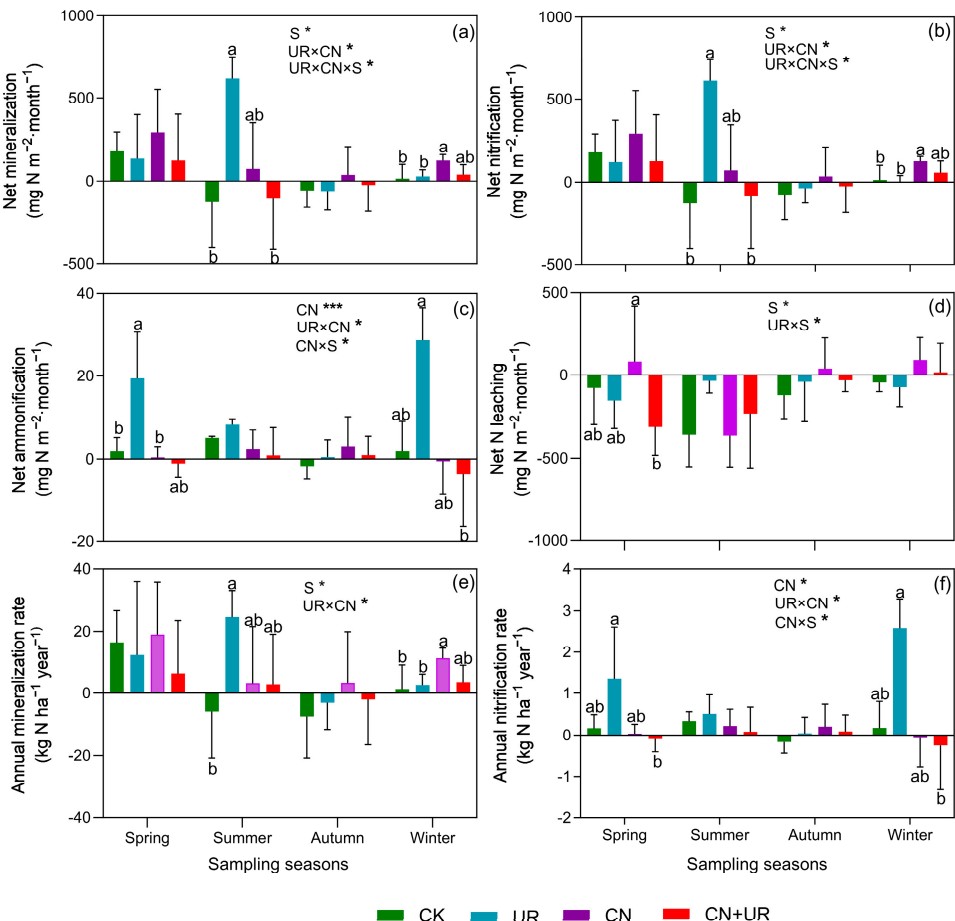

**Figure 3.** Net nitrogen mineralization (**a**), nitrification (**b**), ammonification (**c**), N leaching (**d**), annual mineralization (**e**), and annual nitrification (**f**) in the different applied treatments combined with the seasonal variations in the soil. Data are presented as mean ± SE. Error bars illustrate the variability of the collected data. Different lowercase letters represent significant differences among treatments. The inserted significant values are from three-way ANOVAs. * and *** indicate the significance level of $p < 0.05$ and $p < 0.001$, respectively.

Only the UR treatment exhibited a substantial increase in both net ammonification (mg N m$^{-2}$ month$^{-1}$) and annual nitrification rates (Kg N ha$^{-1}$ year$^{-1}$) (Figure 3c,f), with an average increase of 93.4%, 90.3%, and 38.9% during the winter, spring, and summer, respectively. In contrast, CN as a primary treatment together with its interactions, such as UR × CN and CN × S, was significantly observed for its effects on net ammonification and annual nitrification rates. CN demonstrated an increase in the annual mineralization rate of 19.0 in the spring and 11.3 (Kg N ha$^{-1}$ year$^{-1}$) in the winter, with significant decreases in soil N leaching with CN application in the spring (83.1) and winter (91.6 mg N m$^{-2}$ month$^{-1}$). However, the UR and CN+UR treatments exhibited an increasing trend in N leaching during the spring and winter seasons (Figure 3d,e). The combined treatment of CN + UR showed the least amount of effects on the other N processes. CN exhibited a modestly favorable trend in N transformation processes and N leaching during the autumn season (Figure 3a–f).

### 3.3. Regression Analyses between Soil Factors and N Processes Dynamics

The results of the linear regression scatter plots indicated variations in the significance of the relationships of the N transformation processes with the soil water content (SWC) and the available phosphorus (AP). SWC and net mineralization, nitrification, and annual mineralization rates showed significant negative relationships in the winter ($p = 0.003$) (Figure 4a,b,d). In the spring, a rising trend in SWC revealed a significant positive rela-

tionship with the net ammonification ($p$ = 0.038) (Figure 4c). However, the relationships between SWC, seasonal variations, and other N processes were not statistically significant (Figure S1).

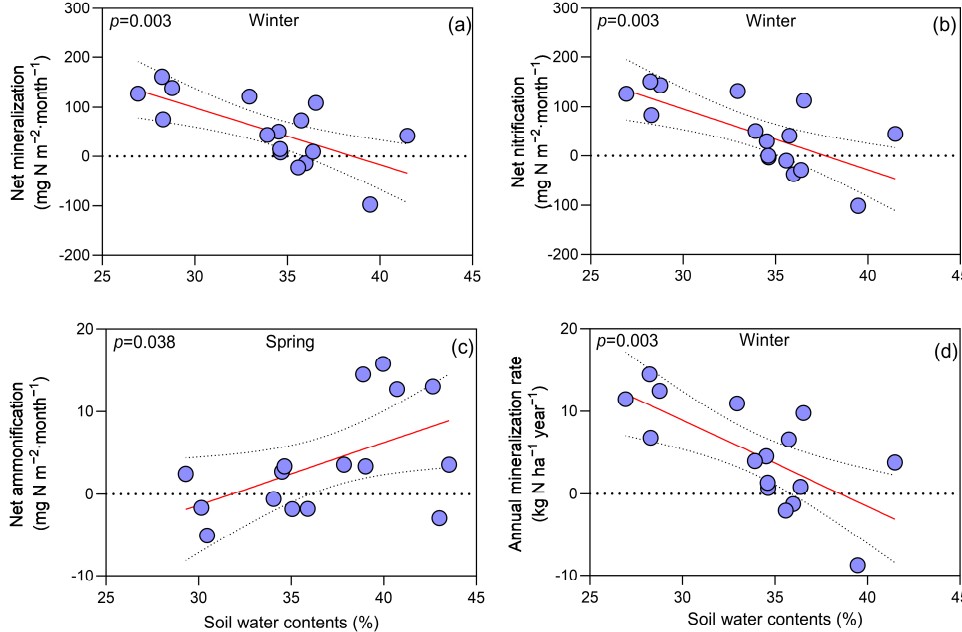

**Figure 4.** Relationships between the soil water content and nitrogen transformation processes: (**a**) net mineralization; (**b**) net nitrification; (**c**) net ammonification; (**d**) annual mineralization rate. The $p$ values are indicated in the figures.

The regression analysis revealed significant relationships between the net ammonification and AP, as well as the annual nitrification rate and AP in the autumn season (Figure 5a,b), while all the other relationships of AP and pH with the remaining N transformation processes were found to be non-significant (Figures S2 and S3).

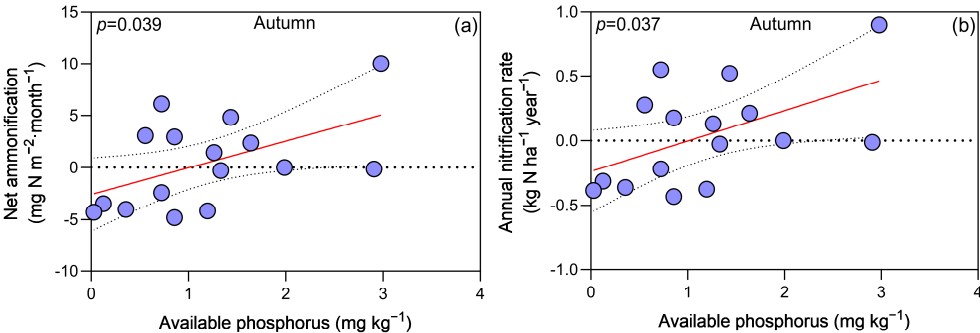

**Figure 5.** Relationships between the availability of phosphorus and nitrogen transformation processes: (**a**) net ammonification; (**b**) annual nitrification rate. The $p$ values are indicated in the figures.

## 4. Discussion

### 4.1. Seasonal Dynamics and the Impact of CN on N Processes

In this study, the effect of seasonality emerged as a key factor that influences net N mineralization and nitrification rates. Previous studies have highlighted that the addition of N in both the summer and autumn leads to a remarkable increase in mineralization processes, highlighting the pronounced seasonal effects [41,42]. Our study revealed a significant influence of CN on the N supply processes (net N mineralization and nitrification) in the spring, winter, and autumn (Figure 3a,b). Furthermore, the soil microbiota dominated the soil ecosystem in the autumn compared with the spring [43]. These seasonal dynamics

in soil activity, coupled with the CN application, are crucial for maintaining the N supply through net mineralization and nitrification rates (Figure 3a–f). It is important to note that CN played a dual role in our study by reducing N leaching while increasing the annual mineralization rates in the spring, winter, and autumn, a finding that differs from the results of previous studies (Figure 3d,e). Previous research has linked CN with N leaching due to the N concentrations in tree organs, reflecting increased photosynthetic rates in forests [44,45].

In this study, the winter exhibited a significant negative correlation between the soil water content (SWC) and N processes (net mineralization, nitrification, and annual mineralization rates), while the spring revealed a positive association with net ammonification (Figure 4a–d), consistent with a previous study highlighting SWC as a key factor in N use efficiency [46]. The results presented in this study indicate that SWC had different effects on the N transformation processes in the different seasons. This contradicts the findings of Reichmann et al. [40], where the increased water availability did not show an impact on the net N mineralization rate. Previous research indicates that soil moisture levels are strongly linked to both the soil redox potential [47] and aeration conditions [48]. Therefore, these three soil factors (soil moisture, redox potential, and aeration), in conjunction with seasonal variations, serve as sensitive indicators of N processes. Furthermore, Sardar et al. and Shibata et al. [49,50] emphasized the importance of the underlying mechanisms, establishing a link between increased net ammonification, total N content, and a change in the fungal-to-bacterial ratio, while also noting that the microbial parameters had no substantial impact on net ammonification and nitrification rates in the autumn. Interestingly, our results revealed a distinctive relationship in the autumn, where the available phosphorus strongly influenced the net ammonification and annual nitrification rates, deviating from earlier research results (Figure 5a,b). However, earlier research has emphasized that phosphorus availability led to reduced losses of N leaching and improved N retention within plant soil ecosystems. This was attributed to the stimulation of plant and microbial growth, as well as increased N uptake [51]. These factors could potentially explain the unique relationship observed in our study during the autumn. Therefore, variations in N supply can be attributed to the different responses of the soil water levels and phosphorus availability that promote soil activity, a notion supported by existing literature [52].

### 4.2. Understory Vegetation Removal, Soil Nitrogen Processes, and Seasonal Correlations

Apart from CN and seasonality, understory vegetation removal (UR) in the forest ecosystem was closely related to net mineralization, nitrification, and annual mineralization rates in the summer. In contrast, UR also had notable effects on net ammonification and annual nitrification rates in the spring, autumn, and winter (Figure 3a–f). Our results suggest that UR serves as an important modulator in accelerating these processes and as an indicator of maintaining the soil ecosystem function. Many other studies have reported that UR improves nutrient availability and promotes microbial growth, as shown by previous research near our site [25]. The existing literature suggests that UR can deliver more inorganic nitrogen ($NO_3^-$-N and $NH_4^+$-N) to the soil microbiota [53,54], which indirectly influences these processes.

The UR leads to increased soil temperatures, mainly as a result of the increased solar radiation reaching the forest floor. This increase in the soil temperature intensifies the photodegradation process during decomposition [55,56]. Wang et al. [57] observed a sustained increase in the soil temperature after the removal of understory vegetation in two subtropical forest areas. The effects of UR extend to significant changes in soil nutrient availability due to changes in the soil microclimate and the modulation of ecological processes, such as decomposition [58]. Accelerated decomposition of the remaining detritus in the soil can potentially enhance the availability of soil nutrients, with a significant increase in N uptake identified as a contributing factor, as noted by Zhu et al. [59]. Furthermore, the removal of understory plants improved the water availability in the soil by reducing the water consumption of the understory vegetation [60,61]. During the wet season, Turner

et al. [62] predominantly noted an increased phosphorus concentration within the soil organic matter. Furthermore, Chen et al. [63] revealed a positive linear correlation between the available phosphorus content, the organic matter content, and the soil water content, suggesting a potential explanation for UR in the forest ecosystem. Our study emphasizes that the increase in plant N pools and microbial N transformation, as well as the reduction of soil N leaching in the soil, are dependent on phosphorus availability and SWC. This suggests that AP and SWC play a mediating role in the dynamics of soil N transformation processes during seasonal changes. The interconnected nature of N and phosphorus cycles underscores the need for future research to consider seasonal scenarios, AP levels, and SWC, and their impact on biogeochemical responses in subtropical forests. Given the N-rich nature of the subtropical forest, UR effectively stabilized N and phosphorus availability, ultimately influencing the N transformation processes in the ecosystem. This suggests that previous studies may have underestimated the importance of the changing phosphorus and N supply processes over the seasons.

In terms of the changes in the net ammonification and annual nitrification rates, our results suggest that the available phosphorus in the autumn serves as an important modulator, a factor rarely reported in previous studies. In this study, the addition of CN during the autumn season demonstrated a slightly favorable trend in N transformation processes and N leaching. Moreover, extensive research has shown that AP tends to limit the maximum growth in subtropical forests [64]. Furthermore, the addition of N has been observed to increase the availability of phosphorus by changing the chemical properties of the soil [65]. Interestingly, we also found a significant role for AP and SWC in modulating N transformation processes depending on the seasonal conditions. Therefore, future research should focus on understanding the biogeochemical processes by considering seasonal scenarios, AP levels, SWC, and active nitrifying microbial communities, regarding the role of CN and UR in response to atmospheric N deposition.

## 5. Conclusions

CN and UR have significantly altered the processes of soil N transformation and leaching in the subtropical forest ecosystem. Taking into account the seasonal variation in N processes, CN significantly increased net nitrification and annual N mineralization rates and possibly reduced N leaching in the spring and winter. In the summer, UR had a strong influence on net N mineralization and nitrification rates. In addition, a strong positive relationship was observed between the available phosphorus and the net ammonification rate in the autumn, as well as the annual nitrification. Overall, our results suggest that seasonal variations should be taken into account when measuring the biogeochemical processes in the soil at certain times of the year. To fully understand the seasonal differences in the response of N cycles to atmospheric N deposition, more research is required, including both CN and UR experiments over an extended period. This is particularly important for the stability of ecosystems and soil management practices where various synthetic N and P fertilizers are used actively to increase productivity and growth.

**Supplementary Materials:** The following supporting information can be downloaded at: https://www.mdpi.com/article/10.3390/f15060962/s1. Figure S1: Non-significant correlations between soil water contents (SWC) and N transformation rates across seasons; Figure S2: Insignificant relationships between N transformation rates and phosphorus availability along with seasonality; Figure S3: The non-significant correlations between soil pH and N transformation rates across seasonal changes.

**Author Contributions:** Conceptualization, S.U.; methodology, S.U.; software, S.U. and J.A.S.; validation, S.U., J.A.S. and J.W.; formal analysis, S.U., W.L., F.S., Y.L., H.D. and G.H.; investigation, J.A.S.; resources, J.W.; data curation, S.U. and W.L.; writing—original draft preparation, S.U.; writing—review and editing, S.U., J.A.S. and J.W.; visualization, S.U. and J.A.S.; supervision, J.W.; project administration, J.W.; funding acquisition, J.W. All authors have read and agreed to the published version of the manuscript.

**Funding:** This work was funded by the National Natural Science Foundation of China (Nos. 32201368, 31570444), the Xingdian Scholar Fund of Yunnan Province, and the Double Top University Fund of Yunnan University.

**Data Availability Statement:** The raw data supporting the conclusions of this article will be made available by the authors on request.

**Conflicts of Interest:** The authors declare no conflicts of interests.

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
