# Peer review of "Effects of Canopy Nitrogen Addition and Understory Vegetation Removal on Nitrogen Transformations in a Subtropical Forest"

_forests, doi:10.3390/f15060962_

Round 1
Reviewer 1 Report
Comments and Suggestions for Authors
I found this manuscript very interesting to be published. The author investigated the role of the forest canopy and understory vegetation in some specific processes of the N cycle. This is an important topic for understanding N deposition in forest ecosystems. But I have some concerns that make me to suggest major revisions. I made some question in the PDF attached. But two major concerns are:
1) Experimental design. Need more clarity about the treatments used. Show your treatments in a Table or sketch.
Also include your general statistical model.
The experiment does not look well planned, Except for the contrast (CN) - (CN+UR) your treatments do not allow to make statistical contrasts between treatments. My related concern with this point is about the interpretations and the discussion in the following sections. How to make sure that your results are supported by this statical design. You do not have a factorial experiment.
2) Results: I recommend to double check your calculations in figures. Assuming a soil bulk density of 1 g/ cm3 (or 1000kg/m3) and your soil depth of 10 cm, some of your numbers seem too high or too low. You could present your numbers in mg of N / kg of soil (numerically equivalent to ppm). This would help to compare your numbers with related studies. I might be wrong, but a double check will be very convenient for your manuscript.

Author Response
Response to the reviewer’s comment
Reviewer #1:
I found this manuscript very interesting to be published. The author investigated the role of the forest canopy and understory vegetation in some specific processes of the N cycle. This is an important topic for understanding N deposition in forest ecosystems. But I have some concerns that make me to suggest major revisions. I made some question in the PDF attached. But two major concerns are:
Response: Thank you for your insightful comments on our manuscript. We agree with these and have incorporated your suggestions in our revised manuscript. For better clarifications of the highlighted issues please see the following point to point responses.
Experimental design. Need more clarity about the treatments used. Show your treatments in a Table or sketch.
Response: Thank you for your nice constructive comment. We have added Figure.1 as a conceptual experimental design for clarity.
Please see page number# 4
Also include your general statistical model.
Response: We appreciate your comment. Regarding general statistical model we have used a randomized block design with four treatments (CK, CN, UR, and CN+UR). Furthermore, to find out the significant difference between treatments we used one way-ANOVA. Likewise, three way ANOVA were also employed to test the main and interactive effects of the CN, UR, and season on N transformation cycles.
Please see page number# 5# Line 216-217
The experiment does not look well planned. Except for the contrast (CN) - (CN+UR) your treatments do not allow to make statistical contrasts between treatments. My related concern with this point is about the interpretations and the discussion in the following sections. How to make sure that your results are supported by this statically design. You do not have a factorial experiment.
Response: Thanks for your nice comment. The treatment group in our randomized block design focuses on CN and UR. Within our design blocks, we use combined CN+UR for correctness, validity, and repeatability in separate block. Since the dependent variables in our study mineralization processes (dependent variables) are greatly influenced by CN and UR management, our randomized block design aims were mainly focused on these main variables (Forest management practices CN and UR) along with seasonal changes.
2) Results: I recommend to double check your calculations in figures. Assuming a soil bulk density of 1 g/ cm3 (or 1000kg/m3) and your soil depth of 10 cm, some of your numbers seem too high or too low. You could present your numbers in mg of N / kg of soil (numerically equivalent to ppm). This would help to compare your numbers with related studies. I might be wrong, but a double check will be very convenient for your manuscript.
Response: Thanks for this comment. Together with seasonal variations, we have confirmed and double check a bulk density of 1.06 g/cm3 at a soil depth of 10 cm. Together with the effects of CN and UR, seasonal variation predominates when it comes to these high or low measured values for available nutrients. As a seasonal representation of N processes, net processes or rates usually show seasonal impacts in mg N m-2 month-1, and our study convert the nitrogen nitrification rates from a monthly (mg N m-2 month-1) to an annual basis (kg N ha-1 yr-1, we used the following formula. (We have used mg N m-2 month-1 for net N rates and kg N ha-1 yr-1 for annual N rates).
(3)
A represents the net annual nitrogen nitrification rates (kg N ha-1 yr-1), and Si denotes the monthly nitrogen nitrification or mineralization rates (mg N m-2 month-1) in each season (spring, summer, autumn, and winter). Conversion constants 4 and 1/100 were applied to convert from a season to a year and from mg N m-2 to kg N ha-1, respectively. We calculated the nitrogen nitrification and mineralization rates in each season and then summed them to obtain the annual rates. For annual rates we used kg N ha-1 accordingly.
Please see page number# 4# Line 201-210
ABSTRACT
Line No# 25# CN (2.5g N m-2 yr-1) and UR
Response: Thanks for your positive comment. We have changed CN unit to 25 kg N ha–1 yr-1 as you suggested. Furthermore, each plot has four blocks that stand for CK, CN, UR, and CN+UR detailed in Figure 1.
Please see page number # 1 line 25
Line No# 27# What is the soil depth for this calculations?
Response: Thanks for your comment. Three polyvinyl chloride tubes (15 cm height, 5 cm in diameter) were inserted at a depth of 10 cm in soil for measuring and calculating these data.
Line No# 28# about word potentially?
Response: Thanks for your comment. To improve clarity, we have eliminated this word from our abstract.
Is it solar radiation effects after understory vegetation removal?
Response: Thanks for your comment. Although we agree with you, there are more details to consider. Additional sunlight radiation penetrates the forest bottom when there is UR, soil temperatures rise as a result. When understory vegetation is removed from two subtropical forest regions in previous studies (please see Discussion for more detail), the temperature of the soil rises, which accelerates the process of photodegradation during its decomposition stage. Moreover, due to modifications in the soil microclimate and the regulation of ecological processes like decomposition, UR can have a substantial impact on the availability of nutrients in the soil.
Wang, Q.W.; Pieristè, M.; Liu, C.; Kenta, T.; Robson, T.M.; Kurokawa, H. The contribution of photodegradation to litter decomposition in a temperate forest gap and understorey. New Phytol. 2021, 229, 2625–2636.
INTRODUCTION
Line No# 114# How was taken into account the treatment CN+ UR in your objective?
Response: Thanks for your nice comment. In our randomized block design, the treatment group primarily focuses on CN and UR. CN+UR is utilized for variability, repetition, validity, and accuracy within blocks. We have observed strong effects of CN and UR in our design and our objectives mainly highlighted CN and UR management practices effects on N processes along with seasonal changes. However, following your suggestions, we have included in result section (that the combined treatment of CN + UR showed the least amount of effects on other N processes).
Please see page# 7 # Line 259
Material and methods
Line No# 138-142# What are the reasons to select 25 kg N ha-1 yr-1 critical rates for N deposition have rates of 50 kg or even more? Experimental design. Need more clarity about the treatments used. Show your treatments in a Table or sketch.
Response: Thanks for your nice comment. The rate of 25 kg N ha-1 yr-1 was sufficient for our area, as each plot was divided into four blocks and each block covered an area of 56.25 m2. We have added Figure for clarity of design.
Please see page #4
Line No# 205; Is this factor ok? Is not 4/100?, Probably I am wrong, but please check this. Every season las 3 months and the year has 12 months.
Response: Thanks for your nice comment. We have changed accordingly as you suggested us.
Please see page #5
RESULTS
Line No # 197 page # 4: Do you have negative rates when the soils lost water and aeration?
Response: Thanks for your nice comment. We agreed with you, as Figure 3 illustrates, we have observed few certain negative values following loss of water and aeration.
Line No# 226; Why you start with P if your main objective is for N?
Response: We have changed accordingly as you suggested us.
Please see page #5
Line No# 263 #page #7; Results: I recommend to double check your calculations in figures. Asuming a soil bulk density of 1 g/ cm3 (or 1000kg/m3) and your soil depth of 10 cm,some of your numbers seem too high or too low. You could present your numbers in mg of N / kg of soil (numerically equivalent to ppm). This would help to compare your numbers with related studies.
Response: Thanks for your nice comment. We have double checked our data as you suggested.
Together with seasonal variations, we confirm and double check a bulk density of 1.6 g/cm3 at a soil depth of 10 cm. Together with the effects of CN and UR, seasonal variation predominates when it comes to these high or low values. As a seasonal representation of N processes, net mineralization processes usually show seasonal impacts in mg N m-2 month-1, and our study convert the nitrogen nitrification rates from a monthly (mg N m-2 month-1) to an annual basis (kg N ha-1 yr-1, we used the following formula. (We have used mg N m-2 month-1 for net seasonal rates and kg N ha-1 yr-1 for annual N rates).
DISCUSSION
Discussion#342# page # 9; Did you measure soil temperature?
Response: Thanks for your nice comment. After the understory vegetation was removed (UR), direct solar radiation was observed reaching the forest floor, however we did not measure it. Previous study highlighted it. (Wang et al., [63])
- Wang, Q.W.; Pieristè, M.; Liu, C.; Kenta, T.; Robson, T.M.; Kurokawa, H. The contribution of photodegradation to litter decomposition in a temperate forest gap and understorey. New Phytol. 2021, 229, 2625–2636.

Reviewer 2 Report
Comments and Suggestions for Authors
This paper studies the effect of undergrowth removal and nitrogen (N) addition to tree crowns on the balance of nutrients in forest ecosystems. The combination of nitrogen removal and entry the from undergrowth and into tree crowns (correspondingly) for the saturated with nitrogen subtropical forest has also not been studied. The seasonal influence of these factors in subtropical forests has not been studied. The authors conducted a field experiment with the following variants: (i) CK - control; (ii) CN - crown N addition (2.5 g N m-2 year-1); (iii) UR - canopy vegetation removal; and (iv) CN+UR - crown N addition plus canopy vegetation removal. The measurements were applied to the mineral soil layer (0–10 cm) during four seasons: spring, summer, autumn, and winter. Research objectives: To determine the rate of nitrogen transformation as a function of soil moisture, available phosphorus content, pH, nitrate leaching through experimental variants and seasons.
Abstract. The abstract does not describe the work done clearly enough, too often quantitative changes are mentioned but there is no analysis of the reasons why these particular changes occurred. The authors did not specify where the studies conducted (For example as lines 106-107: The experiment was carried 106 out in a subtropical forest in southern China). If the reasons for the observed changes are not yet clear at this stage of the work, perhaps the authors would suggest whether adding N to tree crowns and/or removing N with undergrowth biomass initiates or alters which processes. Line 31. 93.35%, 90.29%, and 38.88% in winter, spring, and summer, respectively... One decimal point is probably enough: 93.4%, 90.3%, and 38.9%
1. Introduction. There is an interesting and logical introduction. The authors have well formulated the aim and objectives of the study.
2. Materials and Methods.
The section is generally well described, but I would like to clarify some details.
2.1. Study Site
From Line 122. The climate should be characterized more detail. The manuscript indicates that the rate of the studied processes depends on the time of year. How much did temperature and precipitation vary by seasons (winter, spring, summer and autumn) in the studied subtropical monsoon climate?
2.2. Experimental Design and Measurements
Lines 144-145. “For each nitrogen addition block of the canopy, the required amount of NH4NO3 (269g) was dissolved in 10 L of tap water…”
From this information (NH4NO3 (269g) was dissolved in 10 L), it is not possible to check how much was applied per unit area of soil during the experiment. The authors should either add how the dosing of the working solution was carried out in the experimental plots, taking into account the number of irrigations, or leave only the technical description, leaving the reader to rely on the application rate mentioned above (2.5 g N m-2 yr-1).
The amount of nitrogen entering the tree crowns is given in kg N ha–1 yr-1 (see the Introduction, lines 91-92). One dimension of the parameters should be used in the manuscript.
3.Results - are well presented graphically in the manuscript and in the appendix.
4.Discussion. This part of the manuscript describe correlation of the newly obtained results and the literature and earlier obtained author’s findings.
5. Conclusions. Complex interacting processes have been studied and the data were collected for new work to reveal the dynamics of available nitrogen and phosphorus compounds under the combination of different factors in a subtropical forest.
Author Response
Response to the reviewer’s comment
Reviewer #2:
ABSTRACT. The abstract does not describe the work done clearly enough, too often quantitative changes are mentioned but there is no analysis of the reasons why these particular changes occurred. The authors did not specify where the studies conducted (For example as lines 106-107: The experiment was carried out in a subtropical forest in southern China). If the reasons for the observed changes are not yet clear at this stage of the work, perhaps the authors would suggest whether adding N to tree crowns and/or removing N with undergrowth biomass initiates or alters which processes. Line 31. 93.35%, 90.29%, and 38.88% in winter, spring, and summer, respectively... One decimal point is probably enough: 93.4%, 90.3%, and 38.9%
Response: Thank you for your insightful and critical comment on our manuscript. We have taken into account all the suggestion and points for the better clarity and presentation of our revised manuscript. In addition, this experiment aims to clarify the effect of canopy N addition and understory removal management practices effects on soil N transformation cycling. For this reason, we have applied N to the forest canopies by a high pressure sprinkling irrigation spraying system. We have now revised our abstract according to your potential suggestion. We have taken your advice regarding the decimal points in line 31 which have now been corrected.
Please see our revised abstract & please see page # 1; Line # 19-36; and 106
INTRODUCTION. There is an interesting and logical introduction. The authors have well formulated the aim and objectives of the study.
Response: Thanks for your valuable comment and appreciation.
MATERIALS AND METHODS.
The section is generally well described, but I would like to clarify some details.
2.1. Study Site
From Line 122. The climate should be characterized more detail. The manuscript indicates that the rate of the studied processes depends on the time of year. How much did temperature and precipitation vary by seasons (winter, spring, summer and autumn) in the studied subtropical monsoon climate?
Response: Thanks for your comment. We agree and have added some information based on your suggestion. Spring average temperature is 18.5 ° C, with an average precipitation of 220.6 mm; summer average is 27.9 ° C, with an average precipitation of 177.3 mm; autumn average is 21.3 ° C, with an average precipitation of 73.3 mm; and winter average is 12.7 ° C, with an average precipitation of 94.5 mm.
Please see page # 3; Line # 122-126
2.2. Experimental Design and Measurements
Lines 144-145. “For each nitrogen addition block of the canopy, the required amount of NH4NO3 (269g) was dissolved in 10 L of tap water…”
From this information (NH4NO3 (269g) was dissolved in 10 L), it is not possible to check how much was applied per unit area of soil during the experiment. The authors should either add how the dosing of the working solution was carried out in the experimental plots, taking into account the number of irrigations, or leave only the technical description, leaving the reader to rely on the application rate mentioned above (2.5 g N m-2 yr-1).
Response: Thanks. Basically in this experiment we first dissolved the NH4NO3 in 10 L water for canopy nitrogen addition treatment. After dissolving the required amount, we used a high-pressure spraying apparatus system which was placed at the center of each plot. Less than 1 mm of water is deposited for each plot by this 10L water spraying apparatus system, which won't result in irrigation.
The amount of nitrogen entering the tree crowns is given in kg N ha–1 yr-1 (see the Introduction, lines 91-92). One dimension of the parameters should be used in the manuscript.
Response. Thanks for your nice comment. We have used mg N m-2 month-1 for net nitrogen processes rates (seasonally) and kg N ha-1 yr-1 for annual rates) for clarity.
RESULTS - are well presented graphically in the manuscript and in the appendix.
DISCUSSION. This part of the manuscript describe correlation of the newly obtained results and the literature and earlier obtained author’s findings.
Response: Thanks for your comments and appreciating our work
CONCLUSIONS. Complex interacting processes have been studied and the data were collected for new work to reveal the dynamics of available nitrogen and phosphorus compounds under the combination of different factors in a subtropical forest.
Response: Thanks for your comments. Hope our revised manuscript will meet the standard of Forests.